# Neoadjuvant Chemotherapy plus Radical Surgery in Locally Advanced Cervical Cancer: Retrospective Single-Center Study

**DOI:** 10.3390/cancers15215207

**Published:** 2023-10-29

**Authors:** Liliana Mereu, Basilio Pecorino, Martina Ferrara, Venera Tomaselli, Giuseppe Scibilia, Paolo Scollo

**Affiliations:** 1Obstetrics and Gynecology Unit, “G. Rodolico” University Hospital of Catania, CHIRMED Department, University of Catania, 95123 Catania, Italy; 2Maternal and Child Department, Obstetrics and Gynecology, Cannizzaro Hospital, 95100 Catania, Italy; eliopek@gmail.com (B.P.); martinaferra@hotmail.it (M.F.); paolo.scollo@unikore.it (P.S.); 3Maternal and Child Department, University of Enna “Kore”, 94100 Enna, Italy; 4Economics and Business Department, University of Catania, 95129 Catania, Italy; venera.tomaselli@unict.it; 5Obstetrics and Gynecology, “Giovanni Paolo II” Hospital, 97100 Ragusa, Italy; g.scibilia@libero.it

**Keywords:** cervical cancer, neoadjuvant chemotherapy, radical surgery, lymph node response, disease-free survival, overall survival

## Abstract

**Simple Summary:**

NACT has been used in the setting of trials and obtained satisfactory results in LACC but, according to the 2018 ESGO guidelines, neoadjuvant chemotherapy followed by radical surgery is a controversial alternative. Several pretreatment variables have been found to correlate with the clinical outcome of patients treated with NACT plus radical hysterectomy, such as FIGO stage, tumor size, and lymph node status. Further studies are needed on NACT that analyze the role in the different risk subclasses of patients. The aim of the present study was to analyze pathological responses in patients with locally advanced cervical cancer (LACC) who underwent neoadjuvant platinum-based chemotherapy (NACT) followed by radical hysterectomy. Neoadjuvant chemotherapy followed by radical surgery cannot be considered a standard of care in patients with locally advanced cervical cancer, particularly in the subgroup with pre-NACT imaging suspected for LND metastases.

**Abstract:**

Background: Several pretreatment variables have been found to correlate with the clinical outcome of patients treated with NACT plus radical hysterectomy, such as FIGO stage, tumor size, and lymph node status. Methods: A single-center retrospective observational study to evaluate the use of NACT in LACC, particularly in the lymph-node-positive subpopulation. The study, conducted at the Maternal and Child Department of “Cannizzaro Hospital” in Catania, included patients treated between 2009 and 2019. Multivariate analysis was performed to analyze responses to NACT according to clinicopathologic parameters. Kaplan–Meyer disease-free survival (DFS) and overall survival (OS) curves were generated according to different lymph node status subgroups. Results: A total of 151 consecutive patients were enrolled in the study. Significant independent risk factors for response to NACT were preoperative tumor diameter, parametrium involvement, and lymphoma vascular space invasion (LVSI). T initial diameter at NMR was found to be the independent prognostic predictor for general (*p* = 0.024) and lymph node (LND) response (*p* = 0.028). Tumors between 2 and 6 cm have a better response to NACT than tumors > 6 cm, and LVSI absence was an independent prognostic factor for LND response to NACT. Survival DFS and OS curves were significant for positive vs. negative pathologic LND. Conclusions: Neoadjuvant chemotherapy followed by surgery cannot be considered a standard of care in patients with locally advanced cervical cancer, particularly in the subgroup with pre-NACT imaging suspected for LND metastases.

## 1. Introduction

Globally, cervical cancer is the fourth most common female cancer after breast, colorectal, and lung cancer [1] GLOBOCAN 2020 estimated that, worldwide, there were about 604,000 new cases of cervical cancer, with approximately 342,000 deaths annually. About 83% of all new cervical cancer cases and 88% of all deaths occur in low- and middle-income countries [2].

According to the 2009 staging guidelines of the International Federation of Gynecology and Obstetrics (FIGO) [3], locally advanced cervical cancer (LACC) includes Stages IB2 and IIA2–IVA cervical cancers, and the standard of care is brachytherapy after concurrent chemoradiotherapy (CCRT) [4]. Recent meta-analyses of randomized trials have confirmed that this therapeutic approach significantly improves progression-free survival and overall survival compared with definitive radiotherapy alone [5].

However, in low- to medium-income countries, the implementation of standard radiation therapy protocols (for external beam and brachytherapy) has had major limitations, including the availability of basic infrastructure, appropriate technology, trained personnel with expertise, accessibility to patients, and logistics of high volume [6]. Often, such limitations lead to incomplete treatments or prolonged treatment duration, resulting in suboptimal outcomes.

To overcome these limitations, alternative treatments with reduced need for radiation facilities have been proposed. Neoadjuvant chemotherapy followed by radical hysterectomy has been considered an attractive option for specific subgroups of patients in stages IB and II [7].

NACT has been used in the setting of trials and obtained satisfactory results in LACC [8], but according to the 2018 ESGO guidelines, neoadjuvant chemotherapy followed by radical surgery is a controversial alternative. The benefit of tumor reduction with reference to the prognosis has not yet been proven [9].

Many studies have evaluated the role of chemotherapy before surgery. Since cervical cancer has been shown to have good response rates to modern combination chemotherapy regimens, NACT has been hypothesized to (1) downsize bulky tumors, facilitate surgical resection, and improve local control; (2) assist in the reduction of high-risk histopathological features, therefore, avoiding the need for additional adjuvant locoregional therapy and its toxicities without compromising outcomes; (3) potentially reduce the risk of distal failures by acting on micrometastatic disease; and (4) reserve the functions of ovaries and vagina [10,11].

A recent meta-analysis of randomized and nonrandomized observational studies evaluating response rates and long-term outcomes of different histologies of cervical cancer treated with neoadjuvant chemotherapy and surgery reported clinical complete and partial response rates of 21% and 59.2% for squamous cell cancers and 32.2% and 42.9% for nonsquamous cancers [12]. All studies included in the analysis enrolled patients with poor prognostic factors. Several pretreatment variables have been found to correlate with the clinical outcome of patients treated with NACT plus radical hysterectomy, such as the FIGO stage, tumor size, and lymph node status [12]. NACT followed by radical vaginal trachelectomy and abdominal radical trachelectomy or cone has been described for fertility-sparing treatment in patients with tumors >2 cm but with a higher risk of recurrence [13]. Recent ESGO guidelines agree that further studies are needed on NACT that analyze the role in different risk subclasses of patients, and at the moment, NACT in cervical cancer may be used inside clinical trials [13]. The aim of the present study was to assess the clinical and pathological response, DFS, and OS in different subgroups of patients with LACC FIGO 2009 IB–IIB who consecutively underwent neoadjuvant platinum-based chemotherapy (NACT) followed by radical hysterectomy in a single institution between 2009 and 2019.

## 2. Materials and Methods

### 2.1. Study Design and Data Collection

This study is a single-center retrospective observational study.

The women were selected at Maternal and Child Department of “Cannizzaro Hospital” in Catania (Italy) between 2009 and 2019. A total of 167 consecutive patients with cervical cancer with FIGO stages IB2–IIB (2009) were referred for platinum-based NACT, and 151 of these were successively subjected to radical hysterectomy with pelvic lymphadenectomy.

The design analysis, interpretation of the data, drafting, and revisions were approved by Institutional Review Board of the Cannizzaro Hospital (ID 11/2023); conform to the Helsinki Declaration, the Committee on Publication Ethics guideline (http://publicationethics.org , accessed on 1 May 2023), and the Strengthening the Reporting of Observational Studies in Epidemiology (STROBE) Statement; and were validated by the Enhancing the Quality and Transparency of Health Researcher Network (www.equator-network.org, accessed on 1 May 2023). The data collected were anonymized, considering the observational nature of the study. Each patient signed an informed consent to allow data collection for research purposes.

Inclusion criteria were age > 18 years, patients affected by cervical carcinoma of any histological type, FIGO stage IB2–IIB, undergoing neoadjuvant chemotherapy plus radical hysterectomy, and pelvic lymphadenectomy.

Exclusion criteria: absence of preoperative imaging, patients not submitted to surgery, absence of pathological lymph node evaluation, positive or suspected common iliac and/or para-aortic lymph node, demographic data, pretreatment evaluation included history, physical examination, vaginal–pelvic examination, complete blood analysis, chest X-ray, and abdominal–pelvic computed tomography (CT) scan and/or magnetic resonance imaging (MRI) and/or positron emission tomography (PET). Data were collected from a prospective database specific to oncological patients.

Patient characteristics at initial diagnosis (date, age, FIGO stage, histological type, tumor size, tumor diameter, lymph node status), NACT regimen, type of radical hysterectomy, and pathological responses on surgical specimens were reported for each case. Adjuvant treatment and postoperative follow-up periods, data about disease recurrence and treatments, and survival were evaluated.

### 2.2. Treatment and Management

Clinical staging was performed according to the FIGO 2009 criteria and included pelvic examination, blood analysis, abdomen–pelvis computed tomography and/or PET, nuclear magnetic resonance (NMR; drawing a detailed map of the lesion), chest X-ray, intravenous pyelogram, examination under anesthesia, cystoscopy and/or proctoscopy if clinically indicated.

The choice of NACT regimen was left to the attending physician. Paclitaxel/carboplatin therapy was administered 21 days apart, with an intravenous paclitaxel (Aur, Floriana, Malta) dose of 175 mg/m^2^ or a docetaxel (Accord Healthcare, Milano, Italy) dose of 70 mg/m^2^ administered on day 1, and intravenous carboplatin (Accord Healthcare) with area under the curve (AUC) of 6 mg/mL per min also administered on day 1. As a rule, a maximum of 3 courses of treatment were administered to each patient. Paclitaxel/ifosfamide/cisplatin (TIP) was administered 21 days apart, with an intravenous paclitaxel of 175 mg/m^2^ or a docetaxel dose of 70 mg/m^2^ on day 1 and ifosfamide (Baxter Spa, Monselice, Italy) 5 g/m^2^ (plus mesna 5 g/m^2^) on day 1 and cisplatin (Accord Healthcare) 75 mg/m^2^ on day 2 for 3 cycles.

After NACT, patients underwent radical hysterectomy, unless the tumor responded to preoperative treatment with progressive disease (PD), at which time the tumor was upstaged. In cases in which surgery was not possible, CCRT was adopted.

The response was evaluated based on the response evaluation criteria in solid tumor (RECIST) guidelines version 1.0 [14]. Complete response (CR) was defined as complete disappearance of all target lesions. Partial response (PR) was defined as at least 30% reduction in the sum of the longest diameters of the target lesions. Progressive disease (PD) was defined as a greater than 20% increase in the sum of the greatest diameters of the target lesions within 2 months of study entry, or the appearance of any new lesions, and/or unequivocal progression of existing nontarget lesions. Stable disease (SD) was any condition not meeting any of these three criteria.

In accordance with the World Health Organization (WHO) criteria, NACT responders were defined as patients with a short-term response, CR, and/or PR.

Patients enrolled in the study underwent laparotomic radical hysterectomy and lymph node dissection, including lymph nodes within the pelvis (external iliac nodes, internal iliac nodes, common iliac nodes, suprainguinal nodes, parametrial nodes, and obturator nodes), within 4–8 weeks after the last cycle of chemotherapy.

Postoperative chemoradiotherapy was additionally administered in patients with a positive surgical margin at the vaginal stump, positive lymph node, invasion of the cardinal ligament, or evident invasion of the vasculature and radiotherapy, according to Sedlis criteria [15]. Decisions were undertaken by multidisciplinary oncological team.

Patients were periodically followed up with clinical and radiological examinations until March 2023.

### 2.3. Statistical Analysis

With the aim of assessing NACT plus surgery as a suitable option for patients with LACC and preoperative positive pelvic lymph node in FIGO Stage IB2 and IIB patients as primary endpoint, the relationships among tumor (T) response and lymph node (LND) response, respectively, by type of NACT, tumor dimension at NMR, FIGO stage, histology, LND involvement pre-NACT, parametria involvement pre-NACT, pathological stroma cervical invasion, pathological parametria invasion, pathological specimen margins, pathological LND involvement, and pathological vagina involvement were analysed. Afterward, the same T response and LND response by LVSI with all the above variables were assessed employing symmetric measures such as phi, Cramer’s V, Pearson’s chi-square test, and Fisher’s exact test.

In order to estimate disease-free survival (DFS) and overall survival (OS) measured in months, Kaplan–Meier analyses were performed by log-rank (Mantel–Cox), Breslow (generalized Wilcoxon), and Tarone–Ware tests to compare survival and hazard curves. Firstly, respectively, as events, then LND involvement pre-NACT, pathological LND involvement, LND response, and T response were entered as factors to compare subgroups.

Multivariate logistic regression models by enter and stepwise method were specified to test whether the T response was affected by T dimension. At NMR, pathological stroma cervical invasion and LND response by T dimension at NMR, LVSI, pathological parametria involvement, pathological specimen margins, and pathological LND involvement were analysed employing Hosmer and Lemeshow test to measure the causal relationships by *β* regression coefficients. Statistical data analysis was performed using SPSS 29.0 software package.

## 3. Results

Sixteen patients (9.7%) after NACT remained inoperable and required definitive chemoradiation. The study population consisted of 151 consecutive patients affected by cervical cancer FIGO stages IB2–IIB (2009) undergoing platinum-based NACT and radical hysterectomy with pelvic lymphadenectomy. Preoperative patients’ and tumors‘ characteristics are shown in Table 1.

Among the 151 enrolled women, 65 (43%) had preoperative nodal metastases detected by PET or TC. Postoperative anatomic pathological findings are shown in Table 2.

Twenty-nine (19.2%) patients had a pathological complete response with no residual disease, 110 (72.8%) a partial response with a reduction in tumor cervical dimension, and 12 (7.9%) stable disease.

In 65 (43%) patients, preoperative PET revealed a positive pelvic lymph node, and among these, 37 (56.9%) had a complete pathological response after NACT. Seventeen women had pelvic pathological lymph nodes without any preoperative suspicions at instrumental exams.

A total of 111 (73.5%) women underwent adjuvant therapy after surgery: external radiotherapy in 108 cases, brachytherapy in 25 cases, and chemotherapy in 39 cases. Thirty-one patients were lost to follow-up. The mean month of follow-up was 65 months (SD 47).

Recurrence was detected in 32 patients: pelvic recurrence in 18 cases, abdominal recurrence in 7 cases, lymph node recurrence in 4 cases, and pulmonary recurrence in 3 cases; 17 patients with only local recurrence, and 5 patients with only distant recurrence. Four patients died from tumor during follow-up. In the 12 no-responder patients, no recurrence was detected during the follow-up.

The significant independent risk factors for response to NACT were preoperative tumor (T) diameter, parametrium involvement, and LVSI (Table 3).

Cox hazard regression analysis based on multivariate clinicopathologic characteristics was also performed relative to general and lymph nodal response to NACT: T initial diameter at NMR was found to be the independent prognostic predictor for general (*p* = 0.024) and LND response (*p* = 0.028). In the specific, tumors between 2 and 6 cm had a better response to NACT than tumors > 6 cm, and LVSI absence was an independent prognostic factor for LND response to NACT (*p* = 0.000).

General DFS at 36 and 60 months was 76.3% and 60.5%, respectively, and OS at 36 and 60 months was 75.6% and 74.4%, respectively.

The 36-month DFS in subgroups of women with negative or positive LND pre-NACT were 77.9% and 71.9%, respectively; with negative or positive pathological LND negative were 80.4% and 60.7%, respectively; with LND CR, 84.4%; and with SD, 64.2%.

Kaplan–Meyer disease-free survival (DFS) and overall survival (OS) curves were generated according to different lymph node status subgroups: complete vs. stable LND response, preoperative positive vs. negative lymph node, and pathological positive vs. negative lymph node (Figure 1a,b Figure 2a,b and Figure 3a,b).

## 4. Discussion

Neoadjuvant chemotherapy followed by surgery showed favorable survival outcomes compared with radiotherapy in the prechemoradiation era. In the late 1990s, concurrent chemoradiation became the new standard of care, representing a distinctive shift in treatment philosophy.

Since 1999, the standard treatment of locally advanced cervical cancer (LACC) has been pelvic radiation with concurrent cisplatin, with an absolute improvement of 12% in overall survival compared with radiotherapy alone [16,17,18,19]. It is important to investigate better treatment strategies considering that approximately 40% of patients experience recurrence within 5 years.

Neoadjuvant chemotherapy (NACT) followed by radical hysterectomy has been considered an interesting alternative to CCRT for patients with locally advanced cervical cancer, especially for those with Stage IB2–IIB disease [20].

A meta-analysis of six RCTs including 1078 patients with early or locally advanced disease revealed that NACT followed by radical hysterectomy significantly reduced the risk of progression (hazard ratio (HR) = 0.75, 95% confidence interval (CI) = 0.61–0.93, *p* = 0.008) and the risk of death (HR = 0.77, 95% CI = 0.62–0.96, *p* = 0.02) compared with primary radical hysterectomy [21].

More recently, NACT followed by radical surgery resulted in inferior disease-free survival compared with cisplatin-based concomitant chemoradiation in locally advanced cervical cancer [10]. NACT before definitive radiotherapy has been generally perceived as not beneficial or even detrimental because of the greater toxicity of the chemotherapy regimen, higher recurrence rate, inferior PFS, and lower survival rate [22].

Three contemporary prospective randomized studies have compared neoadjuvant chemotherapy followed by surgery with chemoradiation. The final outcomes of one of them have been published, while the results of the EORTC study have been presented as an abstract, and the third study (CSEM 006) is still recruiting [4].

Nevertheless, a subset of patients with Stages IB2 and IIA might achieve equivalent outcomes with neoadjuvant chemotherapy followed by surgery compared with chemoradiation based on the subgroup analyses from the Tata Memorial Hospital study, although these analyses were not adequately powered for noninferiority or equivalence [10].

The recent ESGO ESMO ESP 2023 guidelines for the management of locally advanced cervical cancer [13] report that definitive radiotherapy should include concomitant chemotherapy whenever possible, and image-guided brachytherapy (IGBT) is an essential component of definitive radiotherapy and should not be replaced with an external boost (photon or proton). If brachytherapy is not available, patients should be referred to a center where this can be performed. In the subgroups of TIB3 and TIIA2 (LN-negative) tumors, there is limited evidence to guide the choice between surgical treatment vs. CTRT with IGBT in LN-negative patients with TIB3 and TIIA2 tumors.

The present study describes the tumor response of patients affected by cervical carcinoma stage FIGO 2009 IB2-IB and consecutively treated with NACT and radical surgery with analysis of possible prognostic factors for T and N responses and DFS and OS in different subgroups of patients. Of a total of 167 patients who were candidates for NACT, 9.7% were deemed inoperable and required definitive chemoradiation. This rate is quite low if we compare it with the Tata study [10], in which 25–30% of patients were not suitable for radical surgery, but we have to take into consideration that our population was a very select subgroup of patients with Stage IB–IIB FIGO 2009. Similar rates were described by Chang et al. [23], with 91% of patients with Stage IB–IIA who underwent surgery after NACT.

The most important finding of the current study is that preoperative tumors of >4 cm were the independent pre-NACT factor of a general poor response to NACT, while LIVS and parametrial involvement were characteristics of a poor NACT lymph nodal response.

Our result in terms of general OS at 5 years: 74% is in line with the Tata Memorial Hospital study with a similar population [10].

We did not find that the T and LND pathological NACT responses were closely related to survival and recurrence. Analyzing different subgroups, no significant difference in terms of DFS or OS has been observed in relation to the presence of pretreatment instrumental lymphadenopathy and lymph nodal response to NACT. We found a statistical difference in terms of OS and DFS between populations with persistent pathological positive and negative pelvic lymph nodes after NACT. These data are in line with previous studies that found no rule of NACT in the LACC population with positive or suspicious lymph nodes. Our study also confirms that in the subgroup of patients with only positive pelvic lymph nodes with an operable T tumor, NACT seemed not to improve DFS or OS.

Even 73% of patients who underwent surgery after neoadjuvant chemotherapy needed adjuvant radiation or chemoradiation, resulting in trimodal therapy with its potential toxicity.

Surgical management of cervical cancer underwent a significant revision after the results of the LACC trial became available [24], so the role of chemotherapy in locally advancer cervical cancer patients has to be investigated further, particularly to establish selection criteria to determine subgroups for which this option significantly improves outcomes [25], for example, bulky tumor [26] or nonsquamous histology [27].

Several limitations of our study should be considered for proper data interpretation. First, the retrospective nature of the study can be considered an intrinsic bias. The relatively low number of enrolled women reduces the power of the conclusions, the group analyzed was heterogeneous with different histology and chemotherapy regimens, and some patients also underwent radiotherapy after radical surgery. Nevertheless, this can be considered one series among several others published, but this can contribute to current knowledge about the management of the disease and be added to a pooled analysis in case of a future systematic review.

## 5. Conclusions

Neoadjuvant chemotherapy followed by surgery cannot be considered a standard of care in patients with locally advanced cervical cancer. This approach needs further clinical research to generate robust high-quality evidence, especially for the subsets that might potentially benefit in terms of survival, toxicity, and quality of life against the gold standard treatment of concomitant chemoradiation.

## Figures and Tables

**Figure 1 cancers-15-05207-f001:**
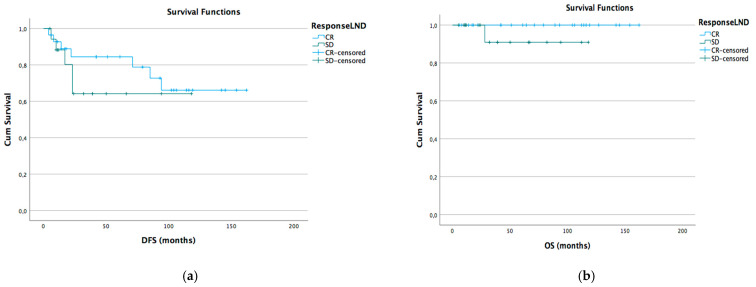
DFS (**a**) and OS (**b**) Kaplan–Meier for LND CR vs. SD (*p* > 0.05).

**Figure 2 cancers-15-05207-f002:**
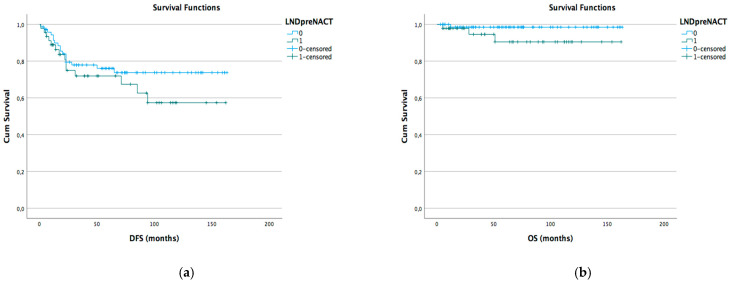
DFS (**a**) and OS (**b**) Kaplan–Meier for preoperative LND positive vs. negative (*p* > 0.05).

**Figure 3 cancers-15-05207-f003:**
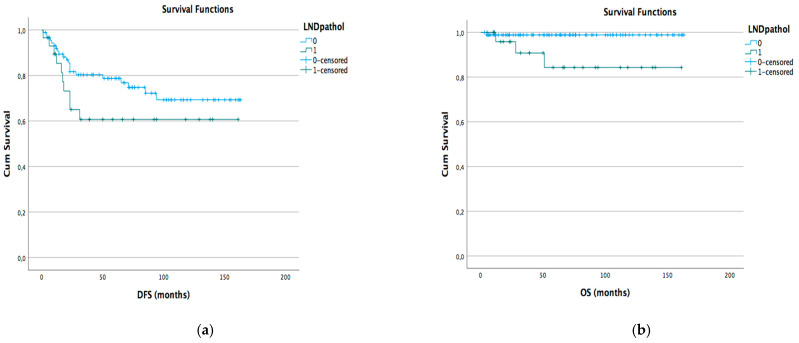
DFS (**a**) and OS (**b**) Kaplan–Meyer for pathological LND positive vs. negative (*p* < 0.05).

**Table 1 cancers-15-05207-t001:** Pre-NACT characteristics.

Characteristic	n
Age, median (range)	53 (24–78)
BMI, median (range)	25 (19–38)
Menopause, n (%)	
Yes	90 (59.6)
No	61 (40.4)
Preoperative FIGO stage, n (%)	
IB2	1 (0.7)
IIA1	2 (1.3)
IIA2	6 (4.0)
IIB	142 (94)
Histology, n (%)	
Squamous	121 (80)
Adenocarcinoma	26 (17.2)
Adenosquamous	4 (2.6)
T dimension at NMR, n (%)	
>2 cm	49 (32.5)
4–6 cm	82 (54.3)
>6 cm	20 (13.2)
Preoperative pelvic LND involvement	
Present	65 (43)
Absent	86 (57)
NACT, n (%)	
CT	27 (17.9)
TIP	122 (80.8)
Other	2 (1.3)
Radical Hysterectomy, n (%)	
Type B	9 (5.9)
Type C	140 (92.7)
Type D	2 (1.3)

**Table 2 cancers-15-05207-t002:** Pathological characteristics.

Parameter	n (%)
T dimension at NMR, n (%)	
0	30 (19.9)
<2 cm	74 (49)
>2–4 cm	38 (25.2)
>4cm	9 (6)
Involvement, n (%)	
Present	30 (19.9)
Absent	121 (80.1)
Vaginal involvement, n (%)	
Present	20 (13.2)
Absent	131 (86.8)
Stroma cervical infiltration, n (%)	
0	30 (19.9)
1/3	26 (17.2)
2/3	25 (16.6)
3/3	70 (46.4)
LIVS, n (%)	
Present	53 (35.1)
Absent	98 (64.9)
Pelvic lymph node involvement, n (%)	
Present	44 (29.1)
Absent	107 (70.9)

**Table 3 cancers-15-05207-t003:** Chemotherapeutic response according to clinicopathologic parameters.

Parameter	n, (%)	General Response (CR + PR)	n (%)	LND Response
Type of NACT CT TIP Other	25 (92.6) 113 (92.6) 1 (50)	0.086	4 (10.8) 33 (89.2) 0 (0)	0.349
FIGO stage (2009) 2 A1 2 A2 2B	2 (100) 6 (100) 128 (91.4)	0.452	1 (50) 2 (100) 34 (55.7)	0.452
Cervical tumor diameter at NMR 2–4 cm 4–6 cm >6 cm	41 (83.7) 78 (95.1) 20 (100)	0.024	11 (68.8) 16 (43.2) 10 (83.3)	**0.028**
Histology AC ASC SCC	23 (88.5) 4 (100) 112 (92.6)	0.655	2 (40) 0 (0) 35 (60.3)	0.173
Lymph node preoperative status Absent Present	78 (90.7) 61 (93.8)	0.579		
LVIS Absent Present	91 (92.9) 48 (90.6)	0.619	34 (91.9) 3 (10.7)	**0.000**
Parametria involvement Absent Present	111 (91.7) 28 (93.3)	0.772	32 (65.3) 5 (31.3)	**0.017**
Vagina involvement Absent Present	119 (90.8) 20 (100)	0.158	35 (57.4) 2 (50)	0.773

## Data Availability

The data presented in this study are available on request from the corresponding author.

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
