# Peer review of "Neoadjuvant Chemotherapy plus Radical Surgery in Locally Advanced Cervical Cancer: Retrospective Single-Center Study"

_cancers, 2023, doi:10.3390/cancers15215207_

Round 1
Reviewer 1 Report
Comments and Suggestions for Authors
This manuscript introduced a retrospective single center study of neoadjuvant chemotherapy. The aim of this study was to analyze the pathological response of patients undergoing radical hysterectomy after neoadjuvant platinum-based chemotherapy (NACT) for Locally Advanced Cervical Cancer (LACC). It turned out that this chemotherapy could not be considered a standard of care for patients with locally advanced cervical cancer, particularly in the subgroup with pre-NACT imaging suspected for LND metastases. This is a meaningful and overall well-written paper that will certainly appeal to many of its readers. However, the following issues should be addressed before the paper is considered suitable for publication in Cancers.
1. There are some format problems in the full text, please check and modify them. For example, many of the same mistakes are repeated on page 3, such as “mg/m2” should be “mg/m2”, “30 %” should be “30%”. In addition, there are other problems, such as page 4, line 152, “ 4-8weeks” should be “ 4-8 weeks”; Table 1, “>6cm” should be “>6 cm”, “>4cm” should be “>4 cm” and the color of “IIB” is red, which is different from other fonts. The same problem appears on pages 7 and 8, such as “p=0.000” and “HR=0.77, 95% CI=0.62-0.96 p=0.02”.
2. Please pay attention to the layout of the article and tables, for example, there is a solid black line at the top of page 7.
3. On Pages 7 and 8, Figure 1, Figure 2 and Figure 3 are not clear, please change them to clear.
4. Please pay attention to the citation format of references, for example, two numbers appear simultaneously like “1. 1.”.
5. It mentioned that “Neoadjuvant chemotherapy followed by surgery can not be considered a standard of care in patients with locally advanced cervical cancer, particularly in the subgroup with pre-NACT imaging suspected for LND metastases” in the article. Meanwhile, the author should also summarize the situations in which NACT is suitable to be used and affirm the significance of the chemotherapy.
6. There are not many references cited in the introduction part of the article, which can be appropriately increased. With the mention of “cervical cancer” and “chemotherapy”, the following published important related papers should be cited: Adv. Mater. 2023, DOI: 10.1002/adma.202304249; Chem. Soc. Rev. 2021, 50, 2839−2891; Exploration 2021, 1, 75–89; VIEW 2022; 3:20200185.
Comments on the Quality of English LanguageModerate editing of English language required
Author Response
We thank you very much for the revision that permitted to improve our paper.
Following our response point to point:
- There are some format problems in the full text, please check and modify them. For example, many of the same mistakes are repeated on page 3, such as “mg/m2” should be “mg/m2”, “30 %” should be “30%”. In addition, there are other problems, such as page 4, line 152, “ 4-8weeks” should be “ 4-8 weeks”; Table 1, “>6cm” should be “>6 cm”, “>4cm” should be “>4 cm” and the color of “IIB” is red, which is different from other fonts. The same problem appears on pages 7 and 8, such as “p=0.000” and “HR=0.77, 95% CI=0.62-0.96 p=0.02”.
We resolved the format problems in the text as suggested
- Please pay attention to the layout of the article and tables, for example, there is a solid black line at the top of page 7.
We restored the correct layout of the tables
- On Pages 7 and 8, Figure 1, Figure 2 and Figure 3 are not clear, please change them to clear.
We improved the quality of the images as suggested
- Please pay attention to the citation format of references, for example, two numbers appear simultaneously like “1. 1.”.
We did a revision and correction of the references
- It mentioned that “Neoadjuvant chemotherapy followed by surgery can not be considered a standard of care in patients with locally advanced cervical cancer, particularly in the subgroup with pre-NACT imaging suspected for LND metastases” in the article. Meanwhile, the author should also summarize the situations in which NACT is suitable to be used and affirm the significance of the chemotherapy.
In the introduction paragraph we specify better the indications for NACT in cervical cancer treatment.
- There are not many references cited in the introduction part of the article, which can be appropriately increased. With the mention of “cervical cancer” and “chemotherapy”, the following published important related papers should be cited: Adv. Mater. 2023, DOI: 10.1002/adma.202304249; Chem. Soc. Rev. 2021, 50, 2839−2891; Exploration 2021, 1, 75–89; VIEW 2022; 3:20200185.
We checked the references suggested but these articles have as topic: new types of chemotherapy in general and not regarding the specific use as neoadjuvant chemotherapy in cervical cancer that is the main focus of these article.
Reviewer 2 Report
Comments and Suggestions for Authors
I reviewed a manuscript entitled “Neo-adjuvant Chemotherapy Plus Radical Surgery in Locally Advanced Cervical Cancer: Retrospective Single Center Study” written by Mereu et. al.
As Authors described in the manuscript, several limitations of the study should be considered for proper data interpretation. First, the retrospective nature of the study can be considered an intrinsic bias. The relatively low number of enrolled women reduces the power of the conclusions, the group analyzed is heterogeneous with different histology and chemotherapy regimens, some patients also underwent radiotherapy after radical surgery. Nevertheless, this can be considered one series among several others published, but this can contribute to current knowledge about the management of the disease and be added to a pooled analysis in case of a future systematic review.
I think it can be accepted by minor revision.
Author Response
Thank you for your comments and suggestions
We revised the manuscript as Suggested.
Sincerely